# Upcycling Chitin Waste and Aged Rice into Fungi Protein Through Fermentation with *Cordyceps militaris*

**DOI:** 10.3390/jof11040315

**Published:** 2025-04-16

**Authors:** Ao Guo, Chunlin Hui, Yongsheng Ma, Xueru Zhang, Lingling Zhang, Shuai Xu, Changtian Li

**Affiliations:** 1Engineering Research Center of Edible and Medicinal Fungi, Ministry of Education, Jilin Agricultural University, Changchun 130118, China; 2Tianjin Institute of Industrial Biotechnology, Chinese Academy of Sciences, Tianjin 300308, China; 3International Joint Research Center for the Creation of New Edible Mushroom Germplasm Resources, Ministry of Science and Technology, Jilin Agricultural University, Changchun 130118, China

**Keywords:** fungi protein, biomass, chitin, FTIR, mushrooms

## Abstract

Microbial protein represents a sustainable alternative to conventional animal protein, yet optimizing substrates for fungal cultivation remains critical. This study demonstrates the successful upcycling of chitin waste and aged rice into fungal protein through fermentation with *Cordyceps militaris*. Substrate formulations (0–20% chitin waste mixed with aged rice) were evaluated for their effects on fungal growth, yield, and metabolite profiles. Results revealed that aged rice alone supported fruiting body yields comparable to fresh rice (9.8 g vs. 9.8 g), with no significant differences in the morphology or growth rate. The addition of 5% chitin waste led to a 17% improvement in yield compared to the control, increasing the average fresh weight of fruiting bodies from 9.8 g to 11.5 g per bottle, while higher chitin levels (20%, T4) suppressed mycelial growth entirely. Fourier-transform infrared spectroscopy (FTIR) and X-ray photoelectron spectroscopy (XPS) confirmed chitin’s structural complexity and nitrogen-rich composition, which slowed the substrate utilization but enriched secondary metabolites. Liquid chromatography–mass spectrometry (LC-MS) identified 1025 metabolites, including up-regulated bioactive compounds (e.g., cordycepin and piplartine) in chitin-amended substrates, linked to amino acid and lipid metabolism pathways. Safety assessments confirmed the absence of toxins, validating the substrates’ suitability for food applications. These findings highlight chitin waste (≤5%) as a viable nitrogen supplement to aged rice, improving the fungal protein yield and bioactive compound synthesis. This approach advances sustainable biomass valorization, offering a scalable strategy to reduce agricultural waste while producing nutrient-dense fungal protein.

## 1. Introduction

With the rapid growth of the global population, global food production is placing unprecedented pressure on the Earth’s ecological systems. Among food categories, conventional protein production—particularly from animal sources—contributes disproportionately to greenhouse gas emissions, land degradation, and water use [1].

By 2050, the environmental pressures of agricultural food production are expected to surpass the Earth’s carrying capacity [2]. This underscores the need for transforming agricultural protein production to promote sustainable development. One such transformation involves replacing red meat in human diets with microbial protein, which can significantly reduce greenhouse gas emissions, land use, freshwater consumption, and the eutrophication of soil and water bodies [3]. Fungal protein is considered a high-quality protein due to its well-balanced amino acid profile, including all essential amino acids [4].

Among these, the branched-chain amino acids (BCAAs)—leucine, isoleucine, and valine—are particularly important as they play critical roles in metabolic and physiological processes [5]. These amino acids cannot be synthesized by the human body, or their synthesis rate is insufficient to meet the body’s needs, requiring their intake through food. Compared to certain plant and animal proteins, fungal protein offers superior nutritional value, as some plant-based proteins contain relatively lower levels of specific essential amino acids, which may affect their overall protein quality. Fungi are versatile organisms that can grow on a wide range of substrates, including agricultural byproducts and food processing waste [6]. This ability makes fungal protein production a sustainable solution, as it can convert otherwise discarded resources into valuable proteins, thus enhancing overall resource utilization. Traditional fungal protein production predominantly relies on liquid fermentation, a method that often leads to the loss of the inherent flavor and taste of the food. As a result, integrating mushroom cultivation into fungal protein production has garnered increasing attention from researchers. Mushrooms not only provide a rich source of protein but also contribute unique flavors and tastes [7].

China is a leading global producer of mushrooms, with its production surpassing 40 million tons in 2022 [8]. Among the various mushroom species, *C. militaris* stands out as an important edible fungus due to its ease of cultivation and high content of bioactive compounds with nutritional and medicinal value [9]. The fruiting body of *C. militaris* is valued for its medicinal and nutritional properties [10]. It is rich in bioactive compounds, including cordycepin, polysaccharides, polyphenols, glycoproteins, superoxide dismutase, and ergosterol, which impart various health benefits, such as anti-tumor, anti-viral, antioxidant, anti-inflammatory, anti-aging, and immunomodulatory effects [11]. Compared to *Cordyceps sinensis*, *C. militaris* offers several advantages, including ease of cultivation, high yield, and the ability to be produced in industrial-scale factories. Recent studies have identified 43 volatile flavor compounds in *C. militaris* food products, including alcohols, aldehydes, acids, ketones, ethers, and olefins, which contribute to its distinctive flavor profile [12].

There are two primary methods for cultivating *C. militaris* [13]. The first involves using natural animals as the culture medium. In this method, *C. militaris* strains are inoculated into healthy living silkworms, tussah larvae, or pupae. The cultures are then maintained under controlled conditions of light, temperature, and humidity for 40 to 50 days to produce *C. militaris* fruiting bodies [14]. While this method results in high-quality fruiting bodies, it is challenging, yields are low, and production costs are high, making it unsuitable for large-scale or factory cultivation. The second method utilizes an artificial solid culture medium, which is a more practical and widely used technique [15]. In this approach, the cultivation medium—typically rice or wheat—is sterilized in bottles before being inoculated with *C. militaris* strains [16]. Under these conditions, mature fruiting bodies can be harvested. This method offers several advantages: it has high yields, uses inexpensive materials, and produces good-quality fruiting bodies, making it well suited for factory-scale cultivation. As knowledge of *C. militaris* increases, there is growing demand for higher yields and improved quality in *C. militaris* products. The choice of cultivation medium plays a significant role in influencing both the yield and quality of the fruiting bodies. For example, when live pupae or larvae are used as substrates, the resulting *C. militaris* fruiting bodies typically have a higher cordycepin content compared to those cultivated on rice or barley [17]. However, this method is more labor-intensive, uses costly raw materials, and yields lower quantities. Thus, there is a need to explore alternative cultivation methods that balance high yield with cost-effective and sustainable practices.

*C. militaris* fruiting bodies typically appear singly or in clusters in the wild, predominantly on the head, thorax, or abdomen of the host [18]. Therefore, utilizing waste materials containing chitin as cultivation substrates may enhance the yield of *C. militaris* while promoting the synthesis of valuable nutrients, such as cordycepin and polysaccharides, in the fruiting bodies. Chitin waste, a typical chitin-rich byproduct, has been shown to serve as an effective substrate for fungal protein production [19]. Fungi, particularly *C. militaris*, secrete a variety of chitinase enzymes and proteases that help break down the substrate during their growth process [20]. Rice, which is processed from paddy grains, loses its outer lemma and cortex, exposing the endosperm. This makes it more vulnerable to adverse environmental factors, such as high temperatures and humidity during the summer, complicating storage [21]. Rice can be classified as either new or old. New rice refers to the current harvest, while aged rice refers to rice stored for over a year. In addition to chitin-rich substrates, *C. militaris* requires starch-rich materials for optimal growth. However, the use of such food sources for cultivation can lead to competition for edible resources, resulting in food losses. To address concerns over food security and minimize waste, cultivating *C. militaris* using non-staple food sources has gained traction. Aged rice, despite its reduced palatability and market value, retains a comparable nutritional content to fresh rice. Its utilization avoids direct competition with human food chains, aligning with the principles of the circular bioeconomy. This study prioritizes aged rice over fresh rice not due to functional superiority, but to demonstrate the feasibility of valorizing low-value agricultural byproducts.

## 2. Materials and Methods

### 2.1. Substrate Preparation and Cultivation

The *C. militaris* strain used in this study was isolated from the fruiting body of commercially available *C. militaris* (https://www.taobao.com/). The solid medium was prepared using common commodity rice purchased from the market (https://www.taobao.com/). Aged rice, which had been stored for approximately two years, was provided by Jilin Agricultural University. Chitin waste, derived from discarded crayfish shells collected from restaurants in Changchun, Jilin Province, was first rinsed thoroughly with running tap water to remove surface residues, then dried at 60 °C, crushed using a grinder, and sieved through a 100-mesh filter to obtain uniform fine particles. No chemical treatment was applied prior to use.

Culture containers consisted of 500 mL specialized glass bottles designed for *C. militaris* cultivation. For each bottle, 40 g of dry material was added, with a water content of 60%. The dry materials were weighed and mixed according to the formulation provided in Table 1. Each treatment was conducted with three biological replicates (*n* = 3). The mixture was then poured into the culture bottle, and water was added according to the specified ratio. The bottle was sealed with polyethylene film, secured with a rubber band, and sterilized at 121 °C for 60 min. Once the culture medium had cooled, 8–10% (*w*/*w*) of the fungi inoculum was added to ensure even distribution on the surface of the medium.

The culture bottles were placed in a dark environment at 18–22 °C. When the white mycelium had grown and fully covered the surface of the medium, the culture entered the initial stage of fruiting body development, characterized by pigmentation of the mycelium and early morphological changes. The cultivation environment was maintained at 25 ± 1 °C with diffuse light at an intensity of approximately 800–1000 lux (measured using a digital lux meter), under a 12 h light/12 h dark cycle. Relative humidity was maintained at 70–80% using a household ultrasonic humidifier (Midea SC-3K15A, Midea Group, Foshan, China) and monitored with a digital thermo-hygrometer (Xiaomi Mijia, Xiaomi Corporation, Beijing, China; accuracy ±3% RH). Once the primordium differentiated and began to grow, the bottles were transferred to the fruiting area. Using a sterile needle, 6 to 8 air holes were punctured evenly in the polyethylene film covering the bottle. The relative humidity of the air in the culture room was kept between 85% and 95%, the temperature was maintained at 18–22 °C, and sufficient diffuse light was provided. The fruiting bodies were ready for harvest once the tops of the fruiting bodies had swelled and granular asci appeared.

### 2.2. Agricultural Trait Detection

The fruiting bodies were harvested upon reaching commercial maturity, and various agronomic traits were recorded. Yield was expressed as biological efficiency (*BE*), calculated as the ratio of the fresh weight of fruiting bodies to the dry weight of the substrate. Each treatment included three biological replicates (*n* = 3). *BE* values were calculated for each replicate and presented as the mean ± standard deviation:$$BE=\left(\frac{\mathrm{weight}\;\mathrm{of}\;\mathrm{fresh}\;\mathrm{mushrooms}\;\mathrm{harvested}}{\mathrm{substrate}\;\mathrm{dry}\;\mathrm{matter}\;\mathrm{content}}\right)\times 100\%$$

To assess the effects of different substrates on the agronomic traits of the fruiting bodies, the diameter and length of the commercially mature fruiting bodies were measured with a vernier caliper. Additionally, the time at which mycelium fully covered the substrate and the completion of color change were recorded.

### 2.3. Study on Surface Characteristics of Substrate

The infrared spectrum analysis was performed using the Thermo Scientific Nicolet iS20 Fourier transform infrared spectrometer (Thermo Fisher Scientific, Waltham, MA, USA), equipped with a mid-infrared DTGS detector. The instrument operated within a wavenumber range of 4000 to 400 cm^−1^, with a resolution of 4 cm^−1^ and 32 accumulations per scan. The new rice, aged rice, and chitin waste samples were dried, crushed, and passed through a 100-mesh sieve prior to analysis. X-ray photoelectron spectroscopy (XPS) was used to analyze the surface elemental composition of aged rice, new rice, and chitin waste powders using a Thermo Scientific ESCALAB 250Xi instrument (Thermo Fisher Scientific, Waltham, MA, USA). Measurements were performed under high vacuum conditions (chamber pressure of approximately 1 × 10^−9^ mbar). Sample preparation followed the same procedure as for the infrared analysis. The XPS spectra were peak-fitted using Origin 2021 software (OriginLab Corporation, Northampton, MA, USA), with C1s (284.80 eV) used for binding energy calibration.

### 2.4. Safety and Compound Evaluation of Fruiting Bodies

We conducted a safety and nutritional evaluation of fruiting bodies produced by t1 and the high-yielding t2 groups as fungal protein foods. Fresh fruiting bodies were harvested and preserved in liquid nitrogen. Prior to testing, 25 mg ± 1 mg of the sample was placed into a centrifuge tube, immediately followed by the addition of 1000 μL of mixed extraction solvent (methanol–acetonitrile–water = 2:2:1, *v*/*v*/*v*). The mixture was thoroughly vortexed and left to stand at −40 °C for 1 h, followed by centrifugation at 12,000 rpm and 4 °C for 15 min to collect supernatant 1. Supernatant 1 was centrifuged again at 12,000 rpm and 4 °C for 15 min to obtain supernatant 2. For non-polar metabolites, chromatographic separation of target compounds was performed using a Vanquish (Thermo Fisher Scientific, Waltham, MA, USA) ultra-high-performance liquid chromatography system equipped with a Phenomenex Kinetex C18 column (Phenomenex Inc., Torrance, CA, USA; 2.1 mm × 50 mm, 2.6 μm). The mobile phase consisted of water containing 0.01% acetic acid (Phase A) and isopropanol–acetonitrile (1:1, *v*/*v*, Phase B). The sample tray temperature was maintained at 4 °C, and the injection volume was set to 2 μL.

### 2.5. Statistical Analysis

All experimental data were analyzed using one-way analysis of variance (ANOVA) followed by Tukey’s Honestly Significant Difference (HSD) test to compare means among treatments. A significance level of α = 0.05 was used for all tests. Results are expressed as mean ± standard deviation (SD). Statistical analysis was performed using SPSS 26.0 (IBM, Armonk, NY, USA). Different lowercase letters within a column indicate statistically significant differences among treatments (*p* < 0.05, one-way ANOVA followed by Tukey’s HSD test, *n* = 3).

## 3. Results and Discussion

### 3.1. Study on Surface Properties of Substrates

#### 3.1.1. FTIR Analysis

Figure 1 illustrates the FTIR spectra of both new and aged rice. The absorption band around 3400 cm^−1^ corresponds to the stretching vibration of the hydrogen bonds between hydroxyl (-OH) or amine (-NH) groups, which is typically associated with starch, proteins, lipids, and water [22]. The absorption band near 2930 cm^−1^ is attributed to the C-H stretching vibration of methyl (-CH_3_) and methylene (-CH_2_-) groups. The band observed around 1652 cm^−1^ is the C=O stretching vibration of amide (-CONH_2_) groups, which can be ascribed to water, proteins, or a combined effect, potentially influenced by polysaccharide vibrations [23]. The absorption peak at approximately 1462 cm^−1^ corresponds to the C-H bending vibration of methyl groups (-CH_2_). The band at 1158 cm^−1^ is characteristic of the C-O stretching vibration of ether linkages (-C-O-C-). Several characteristic fingerprint peaks are observed around 859 cm^−1^. Although the primary absorption peaks for both fresh and aged rice are similar, their relative intensities and peak shapes differ significantly. Fresh rice exhibits sharper peaks at 3356 cm^−1^, 2930 cm^−1^, and 1652 cm^−1^ compared to aged rice. This variation may be attributed to the aging process, where starch, proteins, and lipids in aged rice undergo complexation, leading to the formation of compounds where carbon–hydrogen bonds predominate [24]. Such transformations contribute to the reduced flavor and aroma of aged rice. Differences in other absorption peaks between fresh and aged rice could also result from variations in processing techniques and production batches [25]. Figure 2 presents the FTIR spectra of chitin waste and chitin, showing characteristic absorption peaks corresponding to functional groups such as -NH_2_, -CH, and -C=O, which suggest the presence of proteins, alkanes, and aldehydes associated with their biochemical composition. Absorption peaks at approximately 3286 cm^−1^, 2922 cm^−1^, and 2851 cm^−1^ are attributed to N-H stretching (from -NH_2_ groups and histamine) and C-H stretching vibrations, appearing as broad bands that suggest the presence of proteins, alkanes, and aldehydes. The peaks at 1660 cm^−1^, 1541 cm^−1^, and 1414 cm^−1^ likely result from C=C stretching and N-H bending in olefins. Previous studies have identified significant peaks in chitosan at 3309 cm^−1^, 1629 cm^−1^, 1383 cm^−1^, 1100 cm^−1^, 2927 cm^−1^, and 2850 cm^−1^ [26]. Since chitin wastes are rich in chitosan, many of the absorption peaks of chitin wastes are similar to those observed in chitosan. It is important to note that the FTIR data provide only preliminary insights into the chemical structure of the substrates. Further validation using complementary techniques, such as elemental analysis, NMR, or chromatography, would be necessary to confirm the presence and roles of these functional groups.

#### 3.1.2. XPS Analysis

The primary elements present in aged rice, new rice, and chitin wastes are carbon (C), nitrogen (N), and oxygen (O) (Figure 3). Notably, chitin waste exhibits a relatively high calcium (Ca) content, which is likely due to the presence of minerals predominantly composed of CaCO_3_ [27]. The XPS spectra were analyzed to determine the binding energy and chemical state, and peak-fitting techniques were applied to obtain the C1s spectra for each sample. As shown in Figure 3 and Figure 4, starch (C_6_H_10_O_5_)_n_ is detectable in both new and aged rice, with C-H bonds appearing at approximately 284.8 eV and C-O-C bonds at 286.5 eV. The peak at 288.2 eV likely corresponds to O-C=O bonds, which can be attributed to the proteins present in rice. The 286.5 eV peak of aged rice is broader than that of new rice, potentially due to the aging process, which may have led to modifications in the rice proteins upon exposure to air [28]. In chitin wastes, the C1s spectrum is primarily associated with chitin (C_8_H_13_O_5_N)_n_, with C-H bonds at around 284.8 eV, C-N bonds at 286.0 eV, and N-C=O bonds at 288.2 eV. A comparison of the XPS data from chitin wastes and rice reveals that the carbon source in chitin wastes differs from that in both new and aged rice. The chemical bonds in chitin are more complex and the nitrogen content is higher than in starch, suggesting that chitin may be less readily accessible for use [29]. Despite differences in the sensory quality, the XPS analysis confirms that aged rice retains key carbon-based chemical structures, supporting its continued suitability as a carbon source for biological fermentation.

### 3.2. C. militaris Fruiting Body Germination

As presented in Table 2, both new rice and aged rice exert a comparable efficacy on the growth rate of *C. militaris* mycelium. However, the growth rate of the mycelium declines as the addition of chitin wastes increases. The color change in the *C. militaris* mycelium occurs more rapidly, which may be attributed to the high metabolic activity of the strain. On both pure rice and aged rice media, the differentiation and germination of the primordia occurred significantly faster compared to the experimental group, with the T4 group showing the slowest development, characterized by fewer and smaller primordia. These results indicate that higher chitin waste supplementation negatively impacts the cultivation rate of *C. militaris* fruiting bodies, with a less pronounced effect at lower levels of chitin waste addition. This phenomenon may be explained by the fact that *C. militaris* mycelium grows more efficiently when the carbon-to-nitrogen (C/N) ratio is relatively high. Since the primary component of chitin wastes, chitin, is more challenging for the mycelium to utilize compared to rice starch, it likely hampers the growth of the mycelium [30].

### 3.3. Agronomic Characteristics of C. militaris Fruiting Body

The cultivation experiment revealed that the Ck, T1, T2, and T3 groups successfully cultivated mature *C. militaris*. As shown in Table 3, different culture media influenced the yield, diameter, length, and other characteristics of the *C. militaris* fruiting bodies. When cultivated on fresh rice (Ck: 9.8 ± 1.2 g) and aged rice (T1: 9.8 ± 1.1 g), no significant differences in yield or morphology were observed (*p* > 0.05), confirming that the nutritional degradation of the aged rice does not impair its efficacy as a carbon source. Importantly, aged rice offers distinct advantages in cost-effectiveness (reduced procurement cost by ~30% compared to fresh rice) and sustainability (repurposing stored surplus grains), which are critical for scalable fungal protein production. Figure 4 demonstrates that the aged rice treatment led to an increased abundance of metabolites (e.g., succinate and fructose), suggesting an enhanced TCA cycle and carbohydrate metabolism. This implies that aged rice remains metabolically active and supports fungal biosynthesis.

### 3.4. Statistical Analysis of LC-MS

A comprehensive LC-MS analysis of the Cordyceps fruiting bodies cultivated on aged rice and chitin waste was conducted in positive ionization mode to investigate the metabolite diversity and differential expression among treatments. The metabolite identification was based on accurate mass matching using a high-resolution mass spectrometry database, with a mass error tolerance of ±10 ppm. A total of 1025 metabolites were identified, categorized into 8 superclasses, 53 classes, and 193 subclasses, highlighting the extensive metabolic diversity supported by this cultivation method. Among the superclasses, fatty acids were the most abundant, accounting for 22.44% (230 metabolites) of the total metabolites (Figure 5a). This was followed by other metabolites (19.32%, 198 metabolites), terpenoids (15.9%, 163 metabolites), shikimates and phenylpropanoids (14.54%, 149 metabolites), and alkaloids (14.24%, 146 metabolites). These metabolite categories are known for their bioactive properties, including antioxidant, anti-inflammatory, and antimicrobial activities, suggesting the potential functional and therapeutic value of the cultivated Cordyceps [31]. A safety assessment was conducted with reference to the method of Ma et al. (2020) [32], in which metabolite profiling was used to evaluate the potential toxicity of waste-derived mushroom products. Our results similarly showed no detectable accumulation of toxic or undesired biological compounds. Details are provided in the Appendix A. Principal component analysis (PCA) was performed to explore the metabolic differences between Cordyceps cultivated on the (AR: A_1, A_2, A_3) and (CW: B_1, B_2, B_3). The abscissa PC [1] and the ordinate PC [2] in the figure represent the scores of the first and second principal components, respectively. Each scatter point represents a sample, and the color and shape of the scatter points indicate different groups. The closer the distribution of the sample points is, the more similar the types and contents of metabolites in the samples are. Conversely, the farther apart the samples are, the greater the difference in their overall metabolic levels. All samples are within the 95% confidence interval (Hotelling’s T-squared ellipse). By observing the PCA score plot of all the samples, the overall distribution trend of the samples can be reflected (Figure 5b). The PCA score plot revealed a distinct clustering of the samples, with group AR and group CW showing a clear separation along the principal components. PC1 and PC2 explained 34.4% and 21.1% of the total variance, respectively, while PC3 accounted for an additional 12.3%. This indicates significant metabolic differences between the two cultivation methods.

The volcano plot highlights the differential expression of the metabolites between the two Cordyceps cultivation substrates (Figure 5c). Significantly up-regulated metabolites are marked in red, while down-regulated metabolites are in blue. Notable up-regulated metabolites include piplartine, fructose, and ornithine, which are associated with various bioactive properties [33]. Similarly, key down-regulated metabolites, such as 3-hydroxyoctanoic acid and pipecolic acid, reflect the substrate’s specific influence on metabolic pathways. The threshold for significance (*p* < 0.05) is denoted by the horizontal dashed line, while a fold change cutoff separates metabolites with biologically relevant expression changes. The distribution shows a higher number of up-regulated metabolites, indicating that the CW substrate induces unique metabolic pathways compared to the AR substrate [34]. Collectively, these results highlight the potential of this cultivation strategy to produce Cordyceps with diverse bioactive metabolites and a robust safety profile, suitable for future applications in functional foods and therapeutics.

### 3.5. Enrichment Analysis of Fruiting Bodies in AR and CW Substrate

The heatmap analysis shows distinct metabolic adaptations between the Cordyceps grown on aged rice and chitin waste (Figure 6). The chitin waste group exhibited an upregulation in amino acid metabolism pathways, such as arginine biosynthesis and phenylalanine metabolism, enhancing precursors for protein and secondary metabolite synthesis [35]. An increased activity in the biosynthesis of secondary metabolites and cofactor biosynthesis pathways supports an elevated cordycepin production. Additionally, the linoleic acid and alpha-linolenic acid metabolism were enriched in the chitin waste substrate, reflecting an enhanced lipid synthesis for structural and signaling functions [36]. These results highlight that the chitin waste substrate boosts key biosynthetic pathways for bioactive and functional compounds.

The KEGG pathway enrichment analysis highlighted significant metabolic adaptations in Cordyceps fruiting bodies cultivated on different substrates, with a focus on pathways related to cordycepin biosynthesis, amino acid metabolism, and cofactor production [37]. The biosynthesis of the secondary metabolites pathway (cmt01110) showed a substantial enrichment, including key intermediates such as adenosine-related compounds (C00022 and C00158), which are precursors for cordycepin synthesis. The chitin waste substrate notably enhanced the levels of these metabolites, likely due to its nitrogen-rich composition, promoting enzymatic activities involved in cordycepin production. Additionally, the amino acid biosynthesis pathway (cmt01230) exhibited increased activity, with metabolites such as L-arginine (C00062) and L-tyrosine (C00082) being significantly up-regulated, indicating an enhanced capacity for protein synthesis and the generation of precursors for secondary metabolites, like alkaloids and phenolics. Furthermore, the biosynthesis of the cofactors pathway (cmt01240) and lipid metabolism, including linoleic acid derivatives, was enriched, supporting essential enzymatic reactions and the production of structural and signaling molecules. These results underscore the critical role of the substrate composition in modulating the metabolic pathways of Cordyceps, optimizing the production of bioactive compounds and functional proteins. Although chitin waste served as a sustainable nitrogen source, the lack of comparison with standard supplements, such as ammonium salts or yeast extract, limits the conclusiveness of its efficacy. Future work should incorporate such controls to better assess the nutritional contribution of chitin waste. Moreover, the scalability of the fermentation strategy warrants further investigation through pilot-scale trials or process modeling.

Figure 6 presents the KEGG heatmap and volcano plot of differentially expressed metabolites between the aged rice (AR) and chitin waste (CW) groups, where A_1, A_2, and A_3 represent fruiting bodies cultivated on the aged rice substrate, and B_1, B_2, and B_3 correspond to those cultivated on the chitin waste substrate. In Figure 6, metabolites with up-regulation (log2 fold change ≥ 1, *p* < 0.05) are represented in red, while metabolites with down-regulation (log2 fold change ≤ –1, *p* < 0.05), with a fold-change threshold of ±2.0, are represented in blue.

## 4. Conclusions

This study highlights the potential of aged rice (a non-staple biomass) and chitin waste as sustainable substrates for cultivating *C. militaris*. While the fresh and aged rice exhibited a comparable efficacy in supporting fungal growth (9.8 g yield), the prioritization of aged rice demonstrates a practical strategy to repurpose agricultural surpluses and reduce food resource competition. The chitin waste (≤5%) further enhanced yields by 17% (11.5 g) and enriched bioactive metabolites, validating its role as a nitrogen supplement. Recent studies have shown that nitrogen-rich organic additives, such as shrimp shell powder and soybean meal, can significantly promote the growth and yield of Cordyceps militaris [38]. These findings are consistent with our results, in which the incorporation of 5% chitin waste led to a notable improvement in the fruiting body yield. Compared with synthetic nitrogen sources, chitin not only provides nutrients but may also serve as a structural or bioactive stimulant to the fungal metabolism, suggesting its dual function in substrate enhancement [39]. The results indicate that aged rice provides a stable carbon source without significantly altering the growth rate, morphology, or yield of the fruiting bodies, while chitin waste, at optimal levels (5%), enhances the production of key bioactive compounds, such as cordycepin and polysaccharides. Chitin waste contributes nitrogen-rich components that stimulate specific biosynthetic pathways, including amino acid metabolism and secondary metabolite synthesis, resulting in an improved protein content and functional compound production. This study demonstrated no detectable accumulation of known toxic compounds, such as mycotoxins or hazardous nitrogenous metabolites, as confirmed by untargeted metabolomic profiling. Detailed metabolite identification results are provided in Appendix A. Additionally, principal component and pathway enrichment analyses demonstrated distinct metabolic adaptations between the fruiting bodies grown on aged rice and those grown on chitin waste. The latter substrate showed a significant upregulation in pathways related to cordycepin biosynthesis, amino acid metabolism, and lipid metabolism, underscoring its role in enhancing the nutritional and therapeutic value of the fungal protein. The findings underscore the importance of the substrate composition in modulating fungal metabolism, paving the way for optimizing *C. militaris* cultivation to achieve higher yields and an enhanced bioactive compound production. This study provides a foundation for integrating agricultural byproducts into fungal protein production systems, contributing to sustainable food solutions and the valorization of underutilized biomass. The approach demonstrated in this study provides a preliminary foundation for resource recycling in fungal protein production. With further process optimization and scale-up trials, the model may be translated into decentralized biorefinery systems utilizing locally available agri-waste.

## Figures and Tables

**Figure 1 jof-11-00315-f001:**
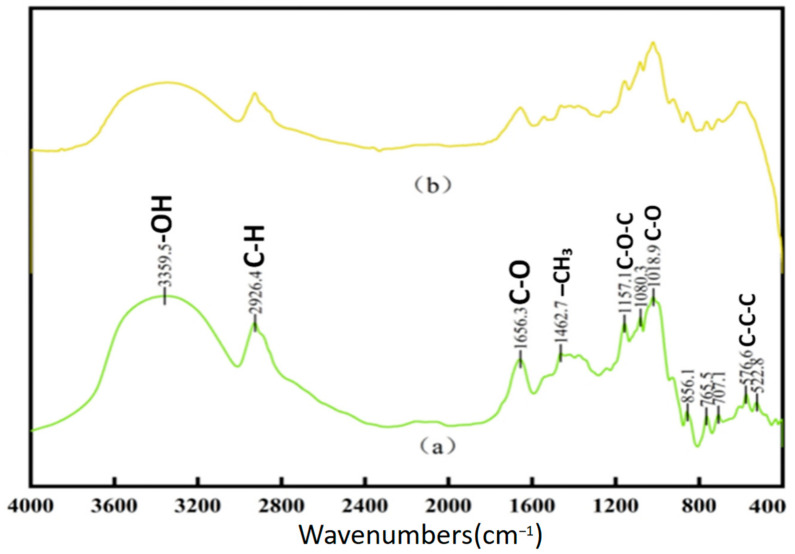
FTIR of new rice (**a**) and aged rice (**b**).

**Figure 2 jof-11-00315-f002:**
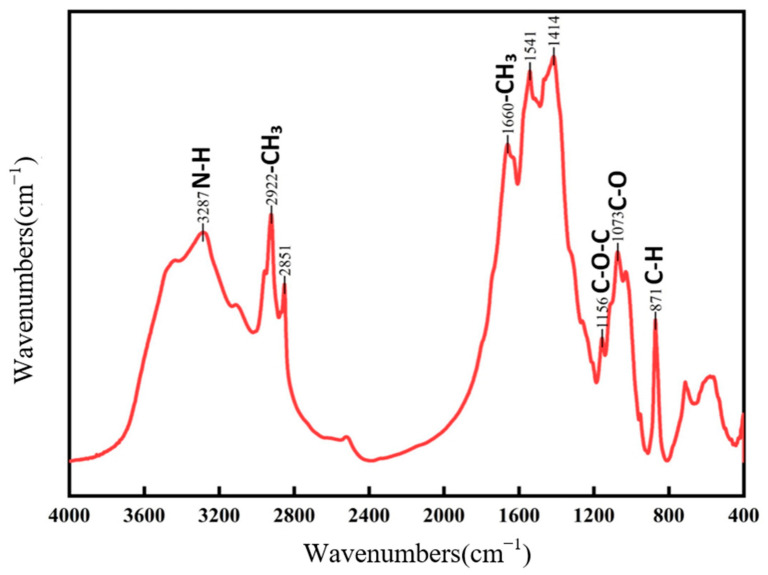
FTIR of chitin waste.

**Figure 3 jof-11-00315-f003:**
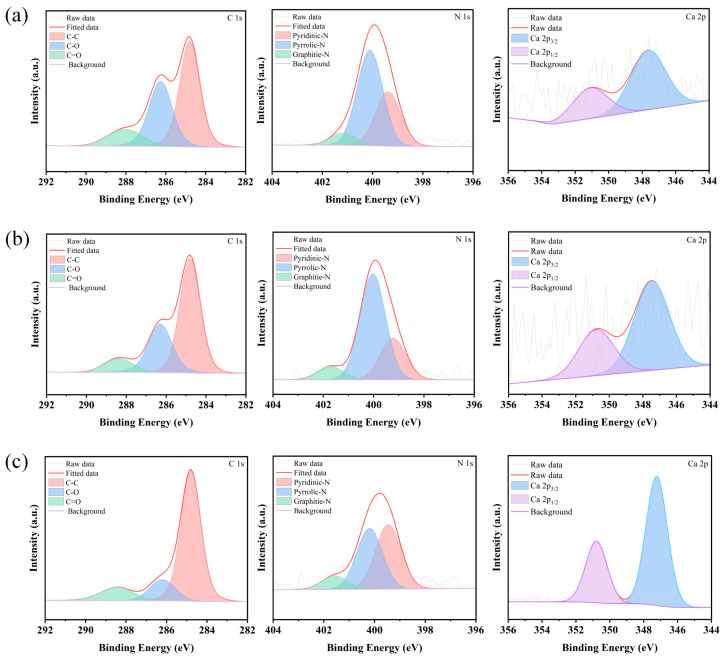
XPS of new rice (**a**), aged rice (**b**), and chitin waste (**c**).

**Figure 4 jof-11-00315-f004:**
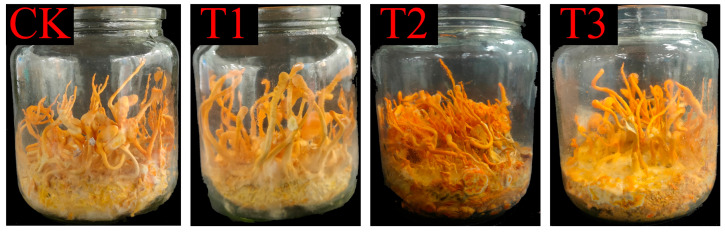
*C. militaris* fruiting body development under different substrate treatments. Each bottle is labeled with its corresponding treatment: T1, T2, T3, and Control. Note variations in cap size and density among treatments.

**Figure 5 jof-11-00315-f005:**
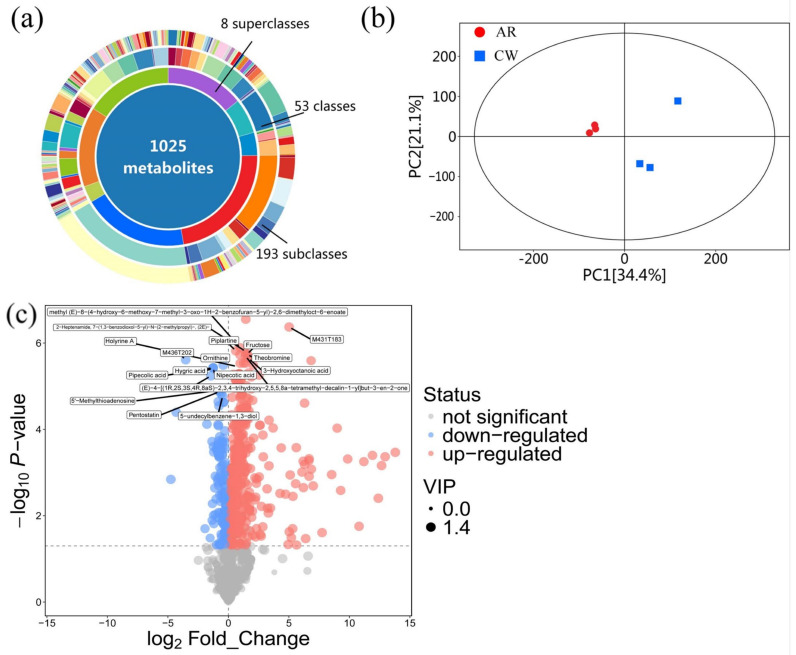
Statistical analysis of LC-MS in aged rice (AR) and chitin waste (CW) groups. (**a**) Hierarchical classification of metabolites into 8 superclasses (highest-level categories), including fatty acids (22.44%), terpenoids (15.9%), and alkaloids (14.24%). (**b**) Further subdivision into 53 classes and 193 subclasses, with a total of 1025 metabolites identified, reflecting the structural and functional diversity of the metabolome. (**c**) Statistical analysis.

**Figure 6 jof-11-00315-f006:**
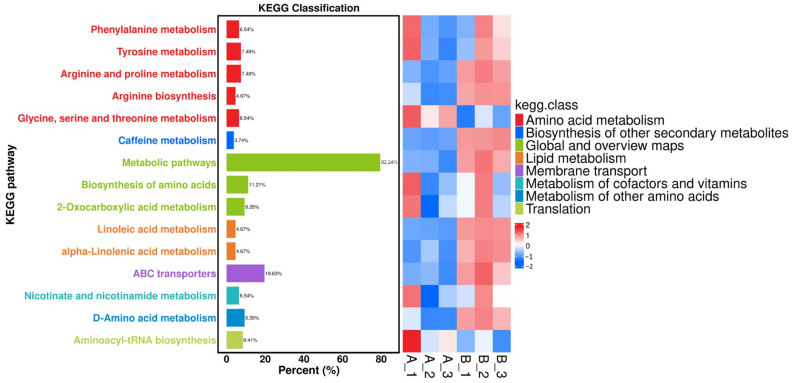
KEGG heatmap of differentially-expressed metabolites between the aged-rice (AR) and chitin-waste (CW) groups. A_1, A_2, and A_3 represent fruiting bodies cultivated on the aged-rice substrate, while B_1, B_2, and B_3 correspond to those cultivated on the chitin-waste substrate. The heatmap shows various metabolic pathways and their classification according to KEGG, with different colors indicating different categories.

**Table 1 jof-11-00315-t001:** *C. militaris* fruiting body culture medium formulation.

Treatment	New Rice	Aged Rice (AR)	Chitin Waste (CW)
Ck	100%	0%	0%
T1	0%	100%	0%
T2	0%	95%	5%
T3	0%	90%	10%
T4	0%	80%	20%

**Table 2 jof-11-00315-t002:** Effects of different culture media on mycelial growth.

Culture Medium	Mycelium Cover Days	Mycelium Full Growth Days/h	Color Change Effect	Primordial Germination Rate%
Ck	5	15	Orange	100
T1	5	15	Orange	100
T2	6	17	Orange	100
T3	8	18	Light yellow	75
T4	13	0	white	25

**Table 3 jof-11-00315-t003:** Effects of different culture media on fruiting body growth.

Culture Medium	Yield/g	Diameter/mm	Length/mm
Ck	9.8 ± 1.2 ^ab^	2.6 ± 0.3 ^ab^	51.2 ± 4.5 ^a^
T1	9.8 ± 1.1 ^ab^	2.8 ± 0.2 ^a^	49.7 ± 5.7 ^b^
T2	11.5 ± 0.2 ^a^	2.7 ± 0.1 ^ab^	46.3 ± 2.3 ^c^
T3	9.1 ± 0.2 ^b^	2.3 ± 0.1 ^b^	46.1 ± 1.6 ^d^
T4	Did not fruiting		

Notes: Values with different superscript letters (a–d) within the same column indicate statistically significant differences (*p* < 0.05) as determined by one-way ANOVA followed by Tukey’s post-hoc test. Groups sharing the same letter are not significantly different.

## Data Availability

The original contributions presented in this study are included in the article and Appendix A. Further inquiries can be directed to the corresponding authors.

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
