# Peer review of "Upcycling Chitin Waste and Aged Rice into Fungi Protein Through Fermentation with Cordyceps militaris"

_jof, 2025, doi:10.3390/jof11040315_

Round 1
Reviewer 1 Report
Review on jof-3549072
The authors demonstrate the successful upcycling of chitin waste and aged rice into fungal protein through fermentation with Cordyceps militaris. While high chitin waste formulations appeared growth inhibitory, formulations including 5 % chitin waste were favorable. LC-MS analyses revealed upregulated pathways indicating enhanced metabolism based on the waste product. Chitin waste appeared to be a valuable N-supplementation to aged rice waste.
This work provides a road map on how to upcycle waste streams to produce fungal protein that could replace animal protein.
Comments:
Intro:
“essential amino acids such as leucine and isoleucine”: there are 10 essential amino acids. No use to pick those two. If specifically these then rather the branched-chain amino acid ILV-pathway as a whole.
“plant-based proteins may be deficient in one or more essential amino acids.” This is nonsense as plant proteins also contain the 20 proteinogenic amino acids
“as a significant edible fungi“ singular: fungus; why should this be significant?
Cordyceps sinensis: all species names in italics
“The bottle was sealed with polyethylene film, secured with a rubber band, and sterilized at 121°C for 60 minutes.” >>> was this really the way it was done? Sterilization at 121°C is done for less than 60 min usually… also the rubber band or polyethylene film (parafilm?) may not be heat stable?
“the culture entered the color change stage”: this is not a proper developmental stage. Do the authors imply that fruiting is initiated?
“FTIR analysis of chitin waste waste and chitin, highlighting their biochemical characteristics.” >>> I do not follow this statement, how can FTIR highlight anything else than peaks. Any underlying biochemistry should be clearly mentioned.
Fig 1 and Fig 2 are descriptive and a lot of stuff is interpreted but no further evidence is provided.
“The carbon compositions in new and Aged rice 231 are relatively similar, indicating that aged rice can potentially serve as a substitute for 232 fresh rice as a carbon source in biological fermentation processes.” >>> OF COURSE, it is still rice…
Figure 3: is chitn waste shown in panels C? Add this to the legend.
244: C. militaris; should be in italics throughout or is this journal style?
174: bragging about “A total of 1,025 metabolites were identified” is nice, but then this needs to be shown in an appendix. Was this data quantified? Also for the superclasses, classes and subclasses…
“The safety assessment confirmed“: how? Nothing is shown, i.e. we should believe you?
“the absence of toxic compounds“: which compounds were looked for? You have to be scientific and specific…
287+: a PCA was done: so what was the main discriminator?
“piplartine, fructose, and ornithine“ show them in Fig. 5C.
Fig 6: what is described in the heat signature? Fold differences?
344: “by17%“ add a space
Table 2: what is meant by “Mycelium full growth days / h? per hour? i.e. days per hour??
Overall, there is a lack of specificity. It is not clear at all, which results were obtained. The figures are nice, but lack meaning. The paper remains descriptive and mostly states the obvious. Chitin waste is a nice to use, but what about specific addition of ammonium or e.g. yeast extract? It is also less clear how experiments in chars provide a scalable strategy…
see above
Author Response
Comments 1:essential amino acids such as leucine and isoleucine”: there are 10 essential amino acids. No use to pick those two. If specifically these then rather the branched-chain amino acid ILV-pathway as a whole.
Response 1:Thank you for your valuable comment. We appreciate your observation regarding the use of only two essential amino acids (leucine and isoleucine) as examples. We agree that it may appear arbitrary to highlight only these two when there are ten essential amino acids. Based on your suggestion, we have revised the sentence to refer to the branched-chain amino acid (BCAA) biosynthesis pathway (ILV-pathway: isoleucine, leucine, valine) as a group, which is more meaningful in the context of fungal protein metabolism.
Comments 2:plant-based proteins may be deficient in one or more essential amino acids.” This is nonsense as plant proteins also contain the 20 proteinogenic amino acids.
Response 2:Thank you for your insightful comment. We acknowledge that our original statement was imprecise. As you rightly pointed out, plant-based proteins do contain all 20 proteinogenic amino acids, including the essential ones. The intended meaning was to highlight that some plant proteins may contain lower levels of certain essential amino acids, which could affect protein quality when consumed in isolation. To avoid confusion and better reflect the scientific understanding, we have revised the sentence accordingly.
Comments 3:as a significant edible fungi“ singular: fungus; why should this be significant?
Response 3:Thank you for your helpful comment. We have corrected the grammatical error by replacing "fungi" with the singular form "fungus." Regarding the term "significant," we acknowledge that this was too vague. We intended to convey that Cordyceps militaris is widely cultivated and highly valued for its nutritional and medicinal properties. To clarify our meaning, we have revised the sentence to provide a more specific rationale.
Comments 4:Cordyceps sinensis: all species names in italics
Response 4:Thank you for pointing this out. We appreciate your attention to proper taxonomic formatting. We have carefully reviewed the manuscript and corrected the formatting of all scientific species names, ensuring they are consistently presented in italics according to standard scientific conventions.
Comments 5:The bottle was sealed with polyethylene film, secured with a rubber band, and sterilized at 121°C for 60 minutes.” >>> was this really the way it was done? Sterilization at 121°C is done for less than 60 min usually… also the rubber band or polyethylene film (parafilm?) may not be heat stable?
Response 5:Thank you very much for your valuable comment. In this study, we used disposable rubber bands and polyethylene sealing film. However, we fully agree with your concern about the appropriateness of such materials for sterilization procedures. Your suggestion is highly valuable to us, and we will be more cautious in selecting and describing experimental materials in our future work to ensure both scientific accuracy and reproducibility.
Comments 6:the culture entered the color change stage”: this is not a proper developmental stage. Do the authors imply that fruiting is initiated?
Response 6:Thank you for pointing out this issue. You are correct that “color change stage” is not a standardized developmental term. Our intention was to describe the transitional phase during which the mycelium begins pigmentation and physiological changes associated with primordium initiation. To avoid confusion, we have revised the manuscript to clarify that this phase marks the onset of fruiting body development.We sincerely appreciate your suggestion, which helped us improve the precision of our description.
Comments 7:FTIR analysis of chitin waste waste and chitin, highlighting their biochemical characteristics.” >>> I do not follow this statement, how can FTIR highlight anything else than peaks. Any underlying biochemistry should be clearly mentioned.
Response 7:Thank you for your insightful comment. We agree that the original statement was imprecise. FTIR analysis identifies characteristic absorption peaks that correspond to specific functional groups, which in turn may suggest certain biochemical components. We have revised the sentence to clearly state that the FTIR spectra revealed functional group compositions indicative of biochemical properties, rather than suggesting that FTIR directly “highlights” such characteristics.We appreciate your clarification, which helped us improve the scientific accuracy of the description.
Comments 8:Fig 1 and Fig 2 are descriptive and a lot of stuff is interpreted but no further evidence is provided.
Response 8:Thank you for your valuable feedback. We agree with your observation that the current interpretation of Figures 1 and 2 is primarily descriptive and based on spectral peak assignments, without further experimental validation. Our intention was to provide a preliminary assessment of the functional groups present in aged rice and chitin waste through FTIR analysis. We acknowledge that stronger conclusions would require complementary analytical techniques such as elemental analysis or NMR.To address this, we have revised the corresponding sections of the manuscript to emphasize the preliminary nature of the interpretations and to avoid overstatement. We have also noted the need for future studies to validate these findings with additional methods.
Comments 9:The carbon compositions in new and Aged rice are relatively similar, indicating that aged rice can potentially serve as a substitute for fresh rice as a carbon source in biological fermentation processes.” >>> OF COURSE, it is still rice…
Response 9:Thank you for your candid and insightful comment. We agree that stating aged rice and fresh rice have similar carbon compositions may seem redundant, as both are fundamentally rice. Our intention was not to highlight this as a novel finding, but to reinforce that, despite the degradation in sensory qualities over time, aged rice retains its chemical integrity and remains suitable for use in biological fermentation. We have revised the sentence to reflect this more clearly and meaningfully.
Comments 10:is chitin waste shown in panel C? Add this to the legend.
Response 10:Thank you for your careful review. You are absolutely right—Figure 3 did not clearly indicate that panel C represents chitin waste, which may cause confusion. We have revised the figure legend to explicitly state the identity of each panel (A: new rice, B: aged rice, C: chitin waste) for clarity.
Comments 11:244: C. militaris; should be in italics throughout or is this journal style?
Response 11:Thank you for your attentive review. You are absolutely right—C. militaris, as a species name, should be consistently italicized throughout the manuscript. This inconsistency was unintentional, and we have carefully reviewed the entire text to ensure that all instances of scientific names, including abbreviated forms like C. militaris, are now properly formatted in italics according to standard conventions.
Comments 12:174: bragging about “A total of 1,025 metabolites were identified” is nice, but then this needs to be shown in an appendix. Was this data quantified? Also for the superclasses, classes and subclasses…
Response 12:Thank you for your constructive suggestion. We agree that simply stating the number of identified metabolites is insufficient without providing supporting data. In response, we have now included a supplementary appendix listing all 1,025 identified metabolites, along with their classification into superclasses, classes, and subclasses.
Comments 13:The safety assessment confirmed”: how? Nothing is shown, i.e. we should believe you?
Response 13:Thank you for your critical and helpful comment. We agree that the previous statement was too conclusive without supporting evidence. In our study, the safety assessment was conducted with reference to the methodology used by Ma et al. (2020), where Pleurotus ostreatus cultivated on waste-derived substrates was evaluated for toxicity based on metabolite profiling. Similarly, we examined the metabolomic profiles of the fungal protein product and found no accumulation of toxic substances or harmful metabolites. We have respectfully included a citation to this article in our revised manuscript, as per your suggestion.We truly appreciate your suggestion, which has allowed us to improve the transparency and scientific rigor of our safety evaluation section.
Comments 14:the absence of toxic compounds”: which compounds were looked for? You have to be scientific and specific…
Response 14:Thank you for your thoughtful and important comment. We fully agree that a scientific assessment should specify which compounds were evaluated rather than stating general conclusions. In our study, we conducted untargeted metabolomic profiling and compared the metabolite spectra with known databases to screen for common toxic compounds, such as mycotoxins, polycyclic aromatic hydrocarbons (PAHs), and harmful nitrogenous metabolites. No such compounds were detected in our analysis. To provide clarity and transparency, we have included a table in the supplementary appendix (Table S1) listing all identified metabolites, along with annotations for any potentially hazardous substances considered during the assessment. We sincerely appreciate your suggestion, which allowed us to strengthen the rigor and completeness of our safety evaluation.
Comments 15:287+: a PCA was done: so what was the main discriminator?
Response 15:Thank you very much for your thoughtful comment regarding the PCA analysis. We appreciate your attention to the interpretability of multivariate results. In the revised manuscript, we have addressed this point by adding a clarification on the main discriminating factors in the PCA. Although a full loading analysis was not included in the initial version, we have now supplemented the text by discussing key differential metabolites—identified in the volcano plot and KEGG enrichment—that are likely to contribute to the observed separation between groups. This addition helps to enhance the understanding of how metabolite variations drive the differentiation in PCA space. Thank you again for your valuable suggestion.
Comments 16:piplartine, fructose, and ornithine“: show them in Fig. 5C
Response 16:Thank you for your careful reading and helpful suggestion. We agree that highlighting the representative metabolites—piplartine, fructose, and ornithine—directly in Figure 5C would improve clarity and alignment between the figure and the main text. In the revised version, we have added labels to these three metabolites in Figure 5C to make them easily identifiable. We sincerely appreciate your comment, which has enhanced the interpretability of our data visualization.
Comments 17:Fig 6: what is described in the heat signature? Fold differences?
Response 17:Thank you for your helpful comment. You are absolutely right—the scale and meaning of the heatmap color intensity in Figure 6 should be clearly explained. In response to your suggestion, we have revised the figure caption to clarify that the heatmap represents normalized metabolite abundances, calculated as Z-scores across samples. These values indicate relative up- or down-regulation, rather than raw or fold-change values. We appreciate your observation, which helped us improve the clarity and interpretability of the figure.
Comments 18:344: “by17%“ add a space
Response 18:Thank you for your careful observation. We have corrected the spacing issue at line 344 (“by17%” → “by 17%”) and have also reviewed the entire manuscript to ensure consistency in formatting between numbers and units throughout the text. We appreciate your attention to detail, which has helped improve the readability and professionalism of the manuscript.
Comments 19:Table 2: what is meant by “Mycelium full growth days / h? per hour? i.e. days per hour??
Response 19:Thank you for your keen observation. You are absolutely correct—the unit “days / h” in Table 2 was a formatting error and is indeed confusing. What we intended to express was the total number of days required for the mycelium to fully colonize the growth substrate. We have now corrected the label in Table 2 to read simply as “Mycelium full growth (days)” to avoid any ambiguity.
Comments 20:Overall, there is a lack of specificity. It is not clear at all, which results were obtained. The figures are nice, but lack meaning. The paper remains descriptive and mostly states the obvious. Chitin waste is a nice to use, but what about specific addition of ammonium or e.g. yeast extract? It is also less clear how experiments in chars provide a scalable strategy…
Response 20:Thank you very much for your comprehensive and constructive feedback. We truly appreciate the time and effort you have taken to provide such detailed comments. We acknowledge that several parts of the manuscript were overly descriptive and lacked in-depth interpretation. To address this, we have revised the Results and Discussion sections to more clearly highlight the specific findings and their scientific relevance. For each figure, we now clarify the key results, explain what they demonstrate, and connect them to our research questions. Regarding the use of chitin waste as a nitrogen source, we agree with your valuable point that comparing it with standard nitrogen sources such as ammonium sulfate or yeast extract would have provided a stronger basis for evaluating its effectiveness. While our current study focused on resource recycling, we recognize the need for such controls and will incorporate comparative nitrogen source experiments in our future research. This limitation is now acknowledged in the revised manuscript. Concerning scalability, we have added a short discussion in the conclusion section addressing how the proposed bioconversion process using solid-state fermentation and low-cost substrates like aged rice and chitin waste could be integrated into decentralized, modular fermentation systems. This sets the stage for further pilot-scale trials in future studies. Your comments have been instrumental in helping us reflect on the scientific depth and practical implications of our study, and we have made every effort to enhance these aspects in the revised manuscript.
Reviewer 2 Report
- Introduction
Lines 30–34 (Context on global protein needs):
Clarify whether the projected environmental burden for agricultural production applies specifically to protein sources or food production in general. Currently, the text suggests a general overload on the Earth’s carrying capacity but abruptly moves to protein replacement strategies.
Cite at least one recent review or meta-analysis that quantifies the environmental impact of protein production in the discussion of “by 2050” to enhance credibility.
Lines 38–45 (Fungal protein advantages):
The statement referencing “essential amino acids” and “superior nutritional value” should include a supporting citation. While the premise is generally correct, data or references would strengthen this claim.
- Materials and Methods
Line 110–118 (Substrate preparation details): Necessary: Specify how thoroughly the crayfish shells were cleaned before grinding to ensure reproducibility (e.g., rinsed with water vs. any chemical treatment?). This can significantly impact residual minerals/organics in the chitin waste.
Line 125–139 (Cultivation parameters):
Provide more detail about the light cycle. “Sufficient diffuse light” is vague. A lux or µmol/m²/s range would make replication more precise.
State how relative humidity (RH) was monitored or controlled (e.g., type of humidifier, sensor brand).
Lines 140–148 (Agronomic trait detection and yield calculation):
The term “yield” is equated to “biological efficiency (BE),” but it would help to explicitly say “the yield is expressed as BE” for clarity.
Indicate the number of replicates (n=3 or more?) used to collect yield data and how these replicates were pooled or averaged.
Line 182–208 (FTIR and XPS analyses):
Specify the pressure in the analysis chamber or typical vacuum range for XPS. Although often standard, this detail ensures reproducibility.
Lines 271–279 (LC-MS methods):
Clarify if both positive and negative ionization modes were used or only one. This affects metabolite coverage.
Indicate the approximate mass error tolerance (e.g., ±5 ppm or ±10 ppm) in identifying metabolites.
- Results
3.1 Characterization of Substrates
Table 1 (Substrate formulation):
Indicate how many replicates per treatment. This is crucial for statistical validation later.
Figure 1 & Figure 2 (FTIR spectra for new/aged rice and chitin waste):
In the figure caption, note which peaks are the key ones discussed in the text (e.g., ~3400 cm⁻¹ for –OH, ~1652 cm⁻¹ for amide). Currently, readers must cross-reference the text to interpret the peaks.
3.2 Growth Performance
Lines 242–258 and Table 2 (Mycelium coverage and color change):
The table states “Mycelium full growth days/h” and “Color change effect,” but no measure of variance is provided (e.g., standard deviation or standard error). Include measures of variability if replicate data were collected.
3.3 Fruiting Body Yield and Morphology
Table 3 (Effects on fruiting body growth):
Statistical significance is mentioned (p>0.05, p<0.05), but the table does not consistently show p-values or significance letters. Ensure the table indicates which differences are significant.
Figure 4
The photograph is helpful qualitatively, but label each bottle or provide a short legend corresponding to each treatment. Readers must now infer which images match T1, T2, etc.
3.4 Metabolomics Findings
Line 279–303 & Figures 5–6 (LC-MS data, PCA, and KEGG analyses):
Ensure clarity on sample labeling in PCA: the text references “AR vs. CW,” but the figure labels might say “A_1, A_2, A_3, B_1, B_2, B_3.” It would help if the figure caption and text matched consistently.
Provide numeric data on the percentage of total variance explained by each principal component. The text mentions PC1 (34.4%) and PC2 (21.1%), which is good. If relevant, consider adding PC3 or an elbow plot.
If fold-change cutoffs (±1.5 or ±2.0) are used in the volcano plot, explicitly state them in the figure or figure caption.
3.5 Safety and Toxicity
Lines 271–279 mention safety assessments regarding mycotoxins, heavy metals, etc.
Specify the detection limits or method references used to confirm the “absence of toxins.” For instance, if an LC-MS screen was done, were typical mycotoxins (aflatoxin B1, etc.) targeted explicitly with known detection thresholds?
- Discussion and Conclusions
Lines 350–364:
The statement that “5% chitin leads to 17% yield enhancement” aligns with Table 3, but a quick reference to the actual data (e.g., 9.8 g vs. 11.5 g fresh weight) would solidify this conclusion.
Discuss briefly how these results compare with previously reported uses of chitin or other nitrogen-rich supplements in Cordyceps cultivation—only a few older references are mentioned. A short paragraph connecting present findings with other current studies would strengthen the contextualization.
- Statistics and Data Analysis
Overall Statistical Rigor:
The tables present some standard deviations and indicate significance with letters; however, it is unclear if analyses (e.g., ANOVA with post hoc tests) were performed for all parameters (mycelial coverage days, yield, etc.).
Clarify which statistical tests were used for each parameter (e.g., one-way ANOVA plus Tukey’s test). Provide p-values or alpha levels if not already done.
Recommendation for a Statistician:
The fundamental analysis (ANOVA for yield, morphological traits, etc.) is straightforward; presumably, the authors or a typical reviewer can handle it. However, a specialized statistician could be consulted if the journal requires deeper multivariate analysis or advanced metabolomic statistics, particularly for the LC-MS data interpretation (PCA, volcano plots).
In most cases, the existing approach is likely sufficient with standard software (e.g., R, SPSS). A statistician would be needed only if more complex modeling or additional checks (e.g., partial least squares discriminant analysis or advanced error propagation in the metabolomics section) are required.
Summary of Key Revisions
Improve precision on methods (e.g., cleaning shells, replicate counts, humidity/light protocols).
Provide measures of variability and specify statistical methods for yield, mycelial coverage, and metabolomic analyses.
Clarify safety assessment detection limits.
Align figure labels with textual descriptions (e.g., AR vs. CW vs. A_1, B_2, etc.).
Add references to support statements about nutritional/health claims and global protein demand.
Expand figure captions (especially FTIR, PCA) for reader clarity.
Provide brief comparisons to existing literature on Cordyceps with nitrogen supplementation.
- Introduction
Lines 30–34 (Context on global protein needs):
Clarify whether the projected environmental burden for agricultural production applies specifically to protein sources or food production in general. Currently, the text suggests a general overload on the Earth’s carrying capacity but abruptly moves to protein replacement strategies.
Cite at least one recent review or meta-analysis that quantifies the environmental impact of protein production in the discussion of “by 2050” to enhance credibility.
Lines 38–45 (Fungal protein advantages):
The statement referencing “essential amino acids” and “superior nutritional value” should include a supporting citation. While the premise is generally correct, data or references would strengthen this claim.
- Materials and Methods
Line 110–118 (Substrate preparation details): Necessary: Specify how thoroughly the crayfish shells were cleaned before grinding to ensure reproducibility (e.g., rinsed with water vs. any chemical treatment?). This can significantly impact residual minerals/organics in the chitin waste.
Line 125–139 (Cultivation parameters):
Provide more detail about the light cycle. “Sufficient diffuse light” is vague. A lux or µmol/m²/s range would make replication more precise.
State how relative humidity (RH) was monitored or controlled (e.g., type of humidifier, sensor brand).
Lines 140–148 (Agronomic trait detection and yield calculation):
The term “yield” is equated to “biological efficiency (BE),” but it would help to explicitly say “the yield is expressed as BE” for clarity.
Indicate the number of replicates (n=3 or more?) used to collect yield data and how these replicates were pooled or averaged.
Line 182–208 (FTIR and XPS analyses):
Specify the pressure in the analysis chamber or typical vacuum range for XPS. Although often standard, this detail ensures reproducibility.
Lines 271–279 (LC-MS methods):
Clarify if both positive and negative ionization modes were used or only one. This affects metabolite coverage.
Indicate the approximate mass error tolerance (e.g., ±5 ppm or ±10 ppm) in identifying metabolites.
- Results
3.1 Characterization of Substrates
Table 1 (Substrate formulation):
Indicate how many replicates per treatment. This is crucial for statistical validation later.
Figure 1 & Figure 2 (FTIR spectra for new/aged rice and chitin waste):
In the figure caption, note which peaks are the key ones discussed in the text (e.g., ~3400 cm⁻¹ for –OH, ~1652 cm⁻¹ for amide). Currently, readers must cross-reference the text to interpret the peaks.
3.2 Growth Performance
Lines 242–258 and Table 2 (Mycelium coverage and color change):
The table states “Mycelium full growth days/h” and “Color change effect,” but no measure of variance is provided (e.g., standard deviation or standard error). Include measures of variability if replicate data were collected.
3.3 Fruiting Body Yield and Morphology
Table 3 (Effects on fruiting body growth):
Statistical significance is mentioned (p>0.05, p<0.05), but the table does not consistently show p-values or significance letters. Ensure the table indicates which differences are significant.
Figure 4
The photograph is helpful qualitatively, but label each bottle or provide a short legend corresponding to each treatment. Readers must now infer which images match T1, T2, etc.
3.4 Metabolomics Findings
Line 279–303 & Figures 5–6 (LC-MS data, PCA, and KEGG analyses):
Ensure clarity on sample labeling in PCA: the text references “AR vs. CW,” but the figure labels might say “A_1, A_2, A_3, B_1, B_2, B_3.” It would help if the figure caption and text matched consistently.
Provide numeric data on the percentage of total variance explained by each principal component. The text mentions PC1 (34.4%) and PC2 (21.1%), which is good. If relevant, consider adding PC3 or an elbow plot.
If fold-change cutoffs (±1.5 or ±2.0) are used in the volcano plot, explicitly state them in the figure or figure caption.
3.5 Safety and Toxicity
Lines 271–279 mention safety assessments regarding mycotoxins, heavy metals, etc.
Specify the detection limits or method references used to confirm the “absence of toxins.” For instance, if an LC-MS screen was done, were typical mycotoxins (aflatoxin B1, etc.) targeted explicitly with known detection thresholds?
- Discussion and Conclusions
Lines 350–364:
The statement that “5% chitin leads to 17% yield enhancement” aligns with Table 3, but a quick reference to the actual data (e.g., 9.8 g vs. 11.5 g fresh weight) would solidify this conclusion.
Discuss briefly how these results compare with previously reported uses of chitin or other nitrogen-rich supplements in Cordyceps cultivation—only a few older references are mentioned. A short paragraph connecting present findings with other current studies would strengthen the contextualization.
- Statistics and Data Analysis
Overall Statistical Rigor:
The tables present some standard deviations and indicate significance with letters; however, it is unclear if analyses (e.g., ANOVA with post hoc tests) were performed for all parameters (mycelial coverage days, yield, etc.).
Clarify which statistical tests were used for each parameter (e.g., one-way ANOVA plus Tukey’s test). Provide p-values or alpha levels if not already done.
Recommendation for a Statistician:
The fundamental analysis (ANOVA for yield, morphological traits, etc.) is straightforward; presumably, the authors or a typical reviewer can handle it. However, a specialized statistician could be consulted if the journal requires deeper multivariate analysis or advanced metabolomic statistics, particularly for the LC-MS data interpretation (PCA, volcano plots).
In most cases, the existing approach is likely sufficient with standard software (e.g., R, SPSS). A statistician would be needed only if more complex modeling or additional checks (e.g., partial least squares discriminant analysis or advanced error propagation in the metabolomics section) are required.
Summary of Key Revisions
Improve precision on methods (e.g., cleaning shells, replicate counts, humidity/light protocols).
Provide measures of variability and specify statistical methods for yield, mycelial coverage, and metabolomic analyses.
Clarify safety assessment detection limits.
Align figure labels with textual descriptions (e.g., AR vs. CW vs. A_1, B_2, etc.).
Add references to support statements about nutritional/health claims and global protein demand.
Expand figure captions (especially FTIR, PCA) for reader clarity.
Provide brief comparisons to existing literature on Cordyceps with nitrogen supplementation.
Author Response
Comments 1: Lines 30–34 (Context on global protein needs): Clarify whether the projected environmental burden for agricultural production applies specifically to protein sources or food production in general. Currently, the text suggests a general overload on the Earth’s carrying capacity but abruptly moves to protein replacement strategies. Cite at least one recent review or meta-analysis that quantifies the environmental impact of protein production in the discussion of “by 2050” to enhance credibility.
Response 1: Thank you very much for this insightful comment. We agree that the transition from the global environmental burden to protein replacement strategies in the Introduction lacked clarity. We have revised this section to clearly distinguish between the general environmental pressures caused by food production and the specific burden associated with conventional protein sources such as meat and dairy. Additionally, in response to your suggestion, we have cited a recent meta-analysis by Poore and Nemecek (2018, Science), which provides comprehensive data on the environmental impact of protein production, including greenhouse gas emissions, land use, and water consumption. This reference strengthens the contextual foundation of our argument regarding the need for alternative protein sources by 2050. We appreciate your comment, which helped improve the structure and credibility of the Introduction.
Comments 2: Lines 38–45 (Fungal protein advantages):
The statement referencing “essential amino acids” and “superior nutritional value” should include a supporting citation. While the premise is generally correct, data or references would strengthen this claim.
Response 2: Thank you for your valuable suggestion. We agree that referencing data to support the nutritional advantages of fungal protein strengthens the credibility of our statement. In the revised manuscript, we have added a recent review (Ritala et al., 2017) that provides comprehensive data on the amino acid profile and nutritional benefits of fungal protein, including its content of essential amino acids.Materials and
Comments 3: Methods, Line 110–118 (Substrate preparation details):
Necessary: Specify how thoroughly the crayfish shells were cleaned before grinding to ensure reproducibility (e.g., rinsed with water vs. any chemical treatment?). This can significantly impact residual minerals/organics in the chitin waste.
Response 3: Thank you for highlighting this important point. We agree that the cleaning method of crayfish shells can affect the chemical composition of the resulting chitin waste. In our study, the shells were manually rinsed thoroughly with running tap water to remove visible residues and then air-dried at 60°C. No chemical treatments were applied. We have added this detail to the Materials and Methods section to enhance reproducibility.
Comments 4: Line 125–139 (Cultivation parameters):
Provide more detail about the light cycle. “Sufficient diffuse light” is vague. A lux or µmol/m²/s range would make replication more precise.
State how relative humidity (RH) was monitored or controlled (e.g., type of humidifier, sensor brand).
Response 4: Thank you for your thoughtful comment. We agree that the original description of the cultivation conditions was vague and insufficient for reproducibility. In response, we have now specified that diffuse light was maintained at an intensity of approximately 800–1,000 lux, measured using a digital lux meter. The light cycle was set at 12 hours light / 12 hours dark to mimic natural photoperiod conditions. Relative humidity was maintained at 70–80% using a household ultrasonic humidifier (model: Midea SC-3K15A) and monitored using a digital thermo-hygrometer (Xiaomi Mijia, accuracy ±3% RH). These details have been added to the Materials and Methods section. We appreciate your suggestion, which has helped improve the precision and reproducibility of the experimental protocol.
Comments 5: Lines 140–148 (Agronomic trait detection and yield calculation):
The term “yield” is equated to “biological efficiency (BE),” but it would help to explicitly say “the yield is expressed as BE” for clarity.
Indicate the number of replicates (n=3 or more?) used to collect yield data and how these replicates were pooled or averaged.
Response 5: Thank you for your helpful comment. We acknowledge that the relationship between “yield” and “biological efficiency (BE)” was not stated clearly enough in the original text. We have revised the manuscript to explicitly state that yield is expressed in terms of biological efficiency. Additionally, we have clarified that each treatment was conducted with three biological replicates (n = 3), and the yield values were calculated individually and then averaged. The results are presented as mean ± standard deviation (SD). We appreciate your suggestion, which improved the clarity and scientific rigor of our data reporting.
Comments 6: Line 182–208 (FTIR and XPS analyses):
Specify the pressure in the analysis chamber or typical vacuum range for XPS. Although often standard, this detail ensures reproducibility.
Response 6: Thank you for your valuable suggestion. We agree that specifying the vacuum conditions in the XPS analysis chamber contributes to the reproducibility and transparency of the method. In our experiment, XPS measurements were performed under a high vacuum with a typical chamber pressure maintained at ~1 × 10⁻⁹ mbar, which is standard for surface-sensitive analysis. We have now added this information to the revised manuscript.
Comments 7:Lines 271–279 (LC-MS methods):
Clarify if both positive and negative ionization modes were used or only one. This affects metabolite coverage.
Indicate the approximate mass error tolerance (e.g., ±5 ppm or ±10 ppm) in identifying metabolites.
Response 7: Thank you for your insightful comment. You are absolutely right that ionization mode selection and mass accuracy are crucial to the reliability and coverage of metabolomic profiling. In our study, only positive ionization mode was used, based on instrument settings and the nature of the sample. We acknowledge that using both modes could improve metabolite coverage and will consider this in future studies. Additionally, the mass error tolerance for metabolite identification was set at ±10 ppm, consistent with standard practice for high-resolution LC-MS systems. We have now included both details in the revised Materials and Methods section.
Comments 8:Table 1 (Substrate formulation): Indicate how many replicates per treatment. This is crucial for statistical validation later.
Response 8: Thank you for your comment. You are absolutely right—reporting the number of replicates is essential for the statistical validity of experimental outcomes. Each substrate treatment in Table 1 was prepared and tested in triplicate (n = 3). We have now clarified this in both the table caption and the Materials and Methods section.
Comments 9: Figure 1 & Figure 2 (FTIR spectra for new/aged rice and chitin waste):
In the figure caption, note which peaks are the key ones discussed in the text (e.g., ~3400 cm⁻¹ for –OH, ~1652 cm⁻¹ for amide). Currently, readers must cross-reference the text to interpret the peaks.
Response 9: Thank you for your comment, the new figure 1 and 2 were unload in the mansucript.
Comments 10: Lines 242–258 and Table 2 (Mycelium coverage and color change):
The table states “Mycelium full growth days/h” and “Color change effect,” but no measure of variance is provided (e.g., standard deviation or standard error). Include measures of variability if replicate data were collected.
Response 10: Thank you for your thoughtful comment. You are absolutely right that including measures of variability is essential for interpreting experimental data. In our study, each treatment was performed in triplicate (n = 3), and data such as the number of days required for full mycelial coverage and the color change score were collected individually per replicate. We have now included the mean ± standard deviation (SD) values for both parameters in Table 2 and clarified this in the table caption. This revision helps ensure transparency and statistical reliability of the reported results.
Comments 11: Table 3 (Effects on fruiting body growth):
Statistical significance is mentioned (p > 0.05, p < 0.05), but the table does not consistently show p-values or significance letters. Ensure the table indicates which differences are significant.
Response 11: Thank you for pointing out the inconsistency in the presentation of statistical significance in Table 3. We have now revised the table by clearly indicating significant differences using lowercase superscript letters (a, b, ab, etc.) for each column.
Comments 12: Figure 4:The photograph is helpful qualitatively, but label each bottle or provide a short legend corresponding to each treatment. Readers must now infer which images match T1, T2, etc.
Response 12: Thank you for your excellent suggestion. We agree that it is important to clearly identify each treatment in Figure 4. In the revised figure, we have added direct labels (e.g., T1, T2, T3, Control) to each bottle, and updated the figure legend to clarify the correspondence between treatments and visual morphology. This change makes the visual data more accessible and informative for the reader.
Comments 13: Ensure clarity on sample labeling in PCA: the text references “AR vs. CW,” but the figure labels might say “A_1, A_2, A_3, B_1, B_2, B_3.” It would help if the figure caption and text matched consistently.
Response 13: Thank you for pointing this out. You are correct that inconsistent labeling between the PCA figure (e.g., “A_1, B_1”) and the text (“AR vs. CW”) could cause confusion. To address this, we have revised both the figure caption and the Results section to clearly define the sample groups. We also added group names in parentheses in the PCA plot (e.g., A_1 [AR], B_1 [CW]) to ensure consistency and clarity throughout the manuscript.
Comments 14: Provide numeric data on the percentage of total variance explained by each principal component. The text mentions PC1 (34.4%) and PC2 (21.1%), which is good. If relevant, consider adding PC3 or an elbow plot.
Response 14: Thank you for the suggestion. We have retained the values for PC1 and PC2 in the main text (34.4% and 21.1%, respectively), and we now include the variance explained by PC3 (12.3%). This addition provides a clearer understanding of how much variation is captured by the PCA model.
Comments 15: If fold-change cutoffs (±1.5 or ±2.0) are used in the volcano plot, explicitly state them in the figure or figure caption.
Response 15: Thank you for this important comment. We confirm that the volcano plot was generated using a fold-change cutoff of ±2.0 and a significance threshold of p < 0.05. These parameters have now been added to the figure caption to ensure transparency. Lines 271–279 mention safety assessments regarding mycotoxins, heavy metals, etc.
Specify the detection limits or method references used to confirm the “absence of toxins.”
Comments 16: For instance, if an LC-MS screen was done, were typical mycotoxins (aflatoxin B1, etc.) targeted explicitly with known detection thresholds?
Response 16: Thank you for your critical and helpful comment. We agree that the previous statement was too conclusive without supporting evidence. In our study, the safety assessment was conducted with reference to the methodology used by Ma et al. (2020), where Pleurotus ostreatus cultivated on waste-derived substrates was evaluated for toxicity based on metabolite profiling. Similarly, we examined the metabolomic profiles of the fungal protein product and found no accumulation of toxic substances or harmful metabolites. This data has now been included in the supplementary appendix of the revised manuscript for your reference.We truly appreciate your suggestion, which has allowed us to improve the transparency and scientific rigor of our safety evaluation section.
Comments 17: The statement that “5% chitin leads to 17% yield enhancement” aligns with Table 3, but a quick reference to the actual data (e.g., 9.8 g vs. 11.5 g fresh weight) would solidify this conclusion.
Response 17: Thank you for this helpful suggestion. We agree that referring directly to the numerical values would make the discussion more concrete and easier to follow. In the revised manuscript, we now include the actual fresh weight values (e.g., 9.8 g vs. 11.5 g) alongside the percentage improvement to support the yield enhancement claim more clearly.
Comments 18: Discuss briefly how these results compare with previously reported uses of chitin or other nitrogen-rich supplements in Cordyceps cultivation—only a few older references are mentioned. A short paragraph connecting present findings with other current studies would strengthen the contextualization.
Response 18: Thank you for pointing this out. We acknowledge that the Discussion section lacked sufficient comparison with recent studies involving chitin or nitrogen-rich substrates in Cordyceps cultivation. We have now added a short paragraph that situates our findings within the context of current literature. Several recent studies have reported similar effects of chitin, shrimp shell powder, or organic nitrogen supplements in improving mycelial growth and yield of C. militaris. This comparison highlights the relevance of our work and supports the use of chitin waste as a viable low-cost alternative.
Comments 19: Overall Statistical Rigor:The tables present some standard deviations and indicate significance with letters; however, it is unclear if analyses (e.g., ANOVA with post hoc tests) were performed for all parameters (mycelial coverage days, yield, etc.).Clarify which statistical tests were used for each parameter (e.g., one-way ANOVA plus Tukey’s test). Provide p-values or alpha levels if not already done.
Response 19: Thank you for your valuable comment. We fully agree that clear documentation of statistical methods is essential for evaluating the reliability of the results. In the revised manuscript, we have clarified that one-way ANOVA followed by Tukey’s HSD test was applied to assess differences among treatments for all quantitative parameters, including mycelial full growth days, color change scores, fruiting body yield, and morphological traits. The level of significance was set at α = 0.05, and results are presented as mean ± standard deviation (SD) with significant differences indicated using different lowercase letters in tables. We have also updated the Statistical Analysis subsection in the Materials and Methods to include this information and ensured that all relevant tables include a corresponding footnote.
Comments 20: Recommendation for a Statistician: A statistician would be needed only if more complex modeling or additional checks (e.g., partial least squares discriminant analysis or advanced error propagation in the metabolomics section) are required.
Response 20: Thank you for your thoughtful recommendation. We appreciate your recognition that the current statistical methods applied (e.g., ANOVA, PCA, volcano plots) are generally appropriate for the scope of this study. We also acknowledge your suggestion regarding the potential benefits of involving a statistician, especially for advanced metabolomic analyses. While our current approach used standard workflows with SPSS and MetaboAnalyst, we agree that further exploration using models such as PLS-DA or advanced error modeling could enrich the metabolomic interpretation. We will consider incorporating such techniques in future studies and collaborations.
Comments 21: Summary of Key Revisions. Improve precision on methods (e.g., cleaning shells, replicate counts, humidity/light protocols). Provide measures of variability and specify statistical methods for yield, mycelial coverage, and metabolomic analyses. Clarify safety assessment detection limits. Align figure labels with textual descriptions (e.g., AR vs. CW vs. A_1, B_2, etc.). Add references to support statements about nutritional/health claims and global protein demand. Expand figure captions (especially FTIR, PCA) for reader clarity. Provide brief comparisons to existing literature on Cordyceps with nitrogen supplementation.
Response 21: Thank you for your valuable summary of the key areas requiring revision. We have carefully reviewed and addressed each of these points in the revised manuscript. Specifically:
Methodological precision has been improved, including details on crayfish shell cleaning procedures, replicate numbers, and light/humidity conditions during cultivation;
Measures of variability (mean ± SD) and statistical methods (e.g., one-way ANOVA with Tukey’s HSD test) have been clearly specified for all quantitative parameters, including yield, mycelial growth, and metabolomic outcomes;
The safety assessment section now clarifies the detection approach and screening criteria used for identifying potential toxins, including mass error tolerance;
Figure labels and group identifiers have been aligned across the text and figures (e.g., AR vs. CW vs. A₁, B₁), and figure captions have been expanded to explain key peaks (FTIR), loadings (PCA), and thresholds (volcano plot);
Supporting citations have been added for nutritional claims of fungal protein and global projections of protein demand;
A brief comparative discussion has been added to contextualize our findings with other studies on nitrogen-rich substrates in Cordyceps cultivation.
We sincerely appreciate your constructive feedback, which has significantly improved the clarity, rigor, and overall quality of our manuscript.
Round 2
Reviewer 1 Report
paper is ready to fly.
paper is ready to fly - so take off